# Towards a Comprehensive Solution for Generating Synthetic Human Mobility Data from Limited Input Data

## Abstract

Human mobility data is essential for transportation planning, urban analytics, and public health, yet access to real-world traces is increasingly restricted due to privacy concerns and institutional controls. Synthetic mobility data offers a viable alternative, but many existing approaches require extensive data inputs or model retraining, raising barriers to reproducibility and privacy assurance. We present a data-minimal framework for generating synthetic human mobility trajectories without additional model training and using only two external sources. The method comprises three conceptual components: (1) profile generation to define agent attributes, (2) agent-based decision-making to plan full-day activity sequences, and (3) route and waypoint visualization to produce structured trajectories and geographic renderings. This design enforces semantic coherence and temporal feasibility while preserving privacy by avoiding real user data. The result is a reproducible, extensible approach for creating synthetic mobility datasets suitable for research and policy analysis under stringent data constraints.

## 1 Introduction

Human mobility data, such as records of trips, trajectories, and activity patterns, forms the backbone of modern transportation research and urban analytics. It is foundational to applications ranging from traffic forecasting and infrastructure planning to epidemiological modeling and disaster response. By capturing how individuals and populations move through space and time, mobility data provides critical evidence for both operational decisions and long-term policy design Barbosa et al. (2018).

Despite its value, access to high-quality mobility data remains severely constrained. Privacy concerns are predominant in this landscape; even after aggregation or pseudonymization, a handful of spatio-temporal points can uniquely identify most individuals, rendering "anonymized" datasets vulnerable to re-identification attacks De Montjoye et al. (2013). These risks reasonably limit data sharing and reuse, creating tension between the need for granular mobility insights and the imperative to protect individual privacy.

Synthetic mobility data has emerged as a promising solution to this dilemma. By generating artificial trajectories that preserve key statistical and behavioral properties of real-world movement, synthetic datasets aim to enable reproducible research and scalable simulations without exposing sensitive information. Recent surveys document a rapidly evolving ecosystem of generative approaches, from rule-based simulators to deep learning models such as GANs, VAEs, and diffusion processes Kapp et al. (2023). However, fidelity remains a multifaceted challenge: matching marginal distributions of trip lengths or departure times is insufficient if higher-order properties, such as activity semantics, network consistency, and temporal coherence, are degraded.

Recent evaluations highlight persistent limitations in state-of-the-art pipelines. One report distinctly shows that even advanced synthetic generation methods, when combined with map matching to real road networks, exhibit fragility: computational failures, implausible route geometries, and unrealistic network-level flows Kapp & Mihaljevic (2023). These findings echo an established observation in movement analytics that map matching itself is a nontrivial, error-prone process Newson & Krumm (2009).

In parallel, large language models (LLMs) have introduced a new paradigm for generative simulation. Prompt-driven "agentic" architectures, which are capable of planning and memory, offer a compelling abstraction for generating synthetic mobility data Park et al. (2023).Prompts can encode daily activity schedules or contextual preferences, while memory mechanisms enforce temporal continuity and diversity. Early demonstrations of such systems in social simulation suggest their potential, yet domain-specific prompting strategies for mobility remain underdeveloped Park et al. (2023); Wang et al. (2024). Traditional agent-based simulators such as MATSim and traffic platforms like SUMO are useful baselines but require significant structured data and lack the generative adaptability and semantic interpretability offered by modern language models Horni et al. (2016); Lopez et al. (2018).

This paper addresses these gaps by introducing a hybrid framework for synthetic human mobility generation. Specifically, we present what we believe to be the first comprehensive demonstration and breakdown of a computational pipeline that generates synthetic human mobility data using only pre-trained models and open-source mapping resources. The framework aligns output trajectories with real road networks, is inherently semantically grounded, and is nearly location-agnostic. The results are qualitatively realistic and may represent an early step toward establishing a new standard for future synthetic data generation approaches.

## 2 BACKGROUND

The study of human mobility has evolved along two complementary lines: *explicit*, rule and agent based simulators that encode behavioral and network constraints, and *implicit*, data-driven generators that learn movement regularities from observations. Platforms such as MATSIM and SUMO provide transparent control over activity schedules, route choice, and network loading, and have been validated across a wide range of planning and operations use cases (Horni et al., 2016; Lopez et al., 2018). In parallel, new literature on deep generative modeling work has sought to synthesize trips and other patterns directly from data, motivated by privacy, reproducibility, and scalability needs. Recent surveys enumerate more than fifty such approaches and catalogue their assumptions and evaluation practices (Kapp et al., 2023; Barbosa et al., 2018). This paper situates our method at that intersection: we retain the discipline of activity and network-constrained modeling while leveraging LLMs for flexibility and coverage.

A few key reassessments between 2023 and 2024 concerned the real-world utility of synthetic mobility under realistic pipelines. One major report called out computational brittleness, jumpy sequences that resist matching, implausible trip lengths, and misallocated flows at intersections despite strong marginal errors (Kapp & Mihaljevic, 2023; Kapp et al., 2023). The HMM formulation by Newson & Krumm 2009 is robust to noise and sparsity, but its accuracy degrades with longer sampling intervals and measurement errors. Broader reviews document similar sensitivities across topological, probabilistic, Kalman, fuzzy-logic, and belief-theoretic map matching families, with later work proposing shortest-path-aided strategies for low-frequency traces (Newson & Krumm, 2009; Quddus et al., 2007; Quddus & Washington, 2015). For synthetic data generation, this means that seemingly minor artifacts in temporal cadence or point geometry can cascade into snapping failures or unrealistic network paths, distorting turn movements among other critical attributes.

Activity semantics offer an additional anchor for realistic data generation. The National Household Travel Survey (NHTS) provides a long-running, nationally representative ontology of trip purposes, activity chains, person/household attributes, vehicle holdings, and temporal patterns. Its 2017 "Summary of Travel Trends" documents methodological updates (address-based sampling, web-based diaries) while retaining constructs foundational to activity-based modeling (Federal Highway Administration; McGuckin & Fucci, 2018). For mobility synthesis, NHTS serves as a principled vocabulary for conditioning agents and day types, and, in our case, for structuring prompts so that generated itineraries reflect accepted definitions in travel behavior research rather than ad hoc labels.

The emergence of LLMs has introduced a complementary abstraction: *prompt-programmed agents* with planning, memory, and reflection. Demonstrations of "generative agents" show that such systems can sustain long-horizon behavior and social interaction (Park et al., 2023). Early adaptations to mobility include LLM-driven travel diary generation from persona and context, and agent frameworks that infer activity patterns before emitting trajectories (Li et al., 2024; Wang et al., 2024;

Bhandari et al., 2024). These prototypes highlight an opportunity for generating richly conditioned day plans as augmentations for ensemble approaches.

Human mobility traces are highly unique. Coarse spatio-temporal data can be enough to re-identify individuals, undermining naive anonymization (De Montjoye et al., 2013). Formal mechanisms extend differential privacy to location information through geo-indistinguishability, and recent work adapts differential privacy to aggregated mobility networks and even to map matching, balancing origin–destination protection with usable network statistics (Andrés et al., 2013; Haydari et al., 2021; 2022).

# 3 METHOD

This approach is intentionally designed to operate under stringent data constraints, incorporating only two external sources and performing no additional model training. This design choice eliminates the possibility of data leakage beyond any inherent in the LLM pretraining corpus, which lies outside the scope of this work. These constraints fundamentally shape the architecture and motivate the methodological decisions described herein.

For clarity of exposition, this method is presented as comprising three conceptual components: (1) profile generation, (2) agent decision-making, and (3) route and waypoint visualization. This segmentation is adopted solely to facilitate explanation and convey the underlying design rationale; it does not imply a prescriptive implementation structure outside of the minimum necessary components.

The process begins with the generation of individual profiles, which define the attributes of synthetic agents. These profiles subsequently inform an agent-based decision making mechanism responsible for planning a full day mobility trace. The resulting sequence of planned locations is then translated into a series of waypoints and rendered through a routing tool to produce both structured trajectory data and a corresponding geographic visualization Figure 1. Collectively, these steps yield a complete synthetic mobility trajectory or multiple for a single day, constructed under strict data minimization constraints.

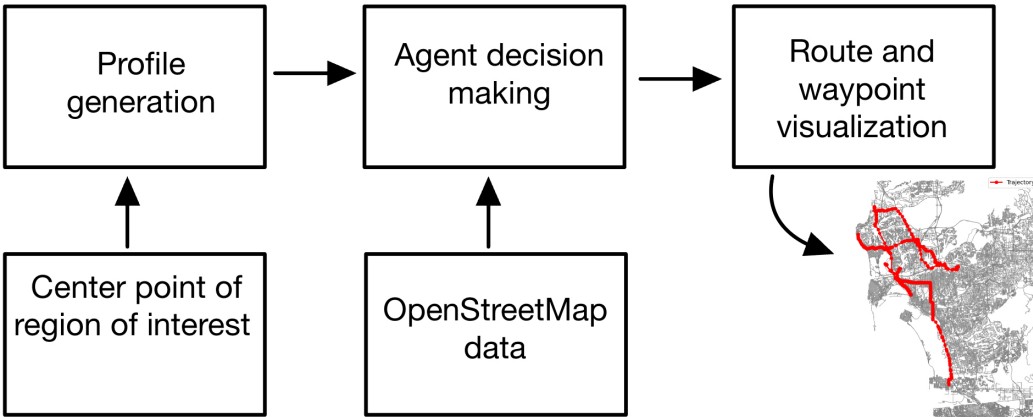

Figure 1: This figure illustrates the flow from the first step to an output of a synthetic mobility trajectory on a map. It starts with an innocuous point of interest as the central region, receives OpenStreetMap information, and then outputs a realistic route matched to the road network for the region.

## 3.1 PROFILE GENERATION

In the first component, the objective is to generate a set of synthetic profiles that represent plausible individuals within a specified geographic region. Only two inputs are required: the region of interest and the number of profiles to produce.

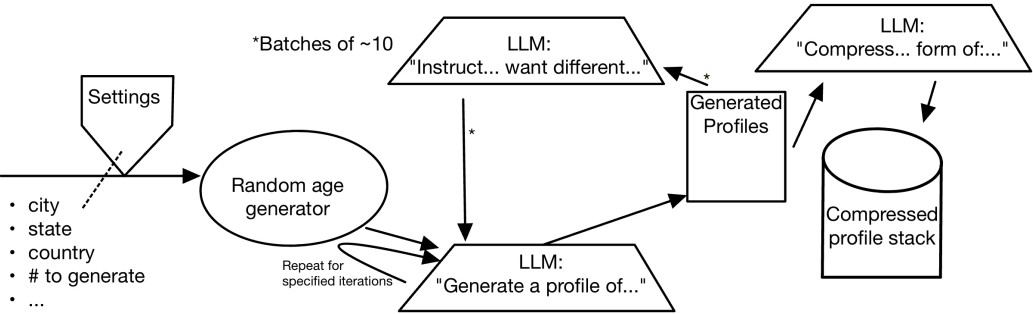

Figure 2: This shows the prompt generation flow used to generate a diverse population set of profiles. The random age generator is used each time the profile generating LLM is queried. The generated profiles are kept temporarily in a batch so that after every 10 another LLM can update a running examination on the diversity of the profiles and suggest professions, for example, that are overrepresented while providing alternative suggestions for the next round of generated profiles. Each generated profile is passed to another LLM to fit the information into a neatly documented text-based form.

To approximate realistic demographic variation, an age value is sampled for each profile from a shifted Beta distribution. This choice reflects the skew typically observed in population age distributions and reduces reliance on the language model for structured numerical attributes, which helps avoid inconsistencies.

Once an age is assigned, a LLM is prompted to generate additional attributes for a plausible individual in the specified region. These attributes provide the behavioral and contextual basis for downstream agent-based decision-making. Including regional context improves relevance, but it remains superficial because no external datasets are incorporated. This is an intentional decision to minimize privacy risks.

Aggregated or auxiliary datasets, even in anonymized form, were excluded at this stage to maintain strict adherence to privacy-preserving principles. The process finishes with an additional language model query for each profile that produces a concise summary of the individual's characteristics in a standardized format for subsequent processing.

## 3.2 AGENT DECISION-MAKING

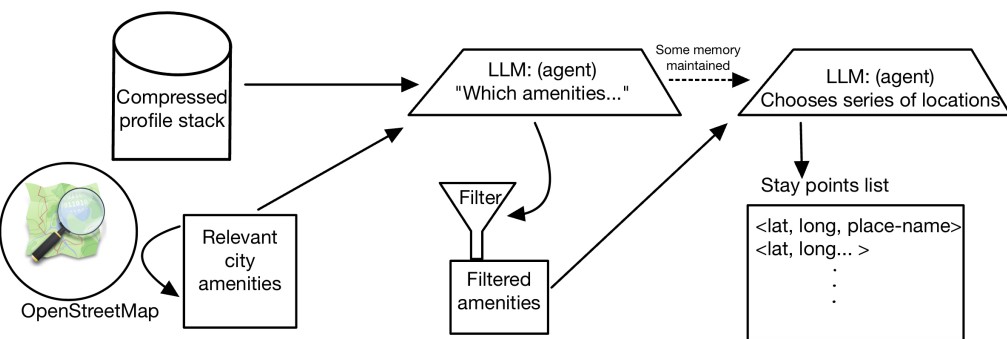

Figure 3: This shows the information flow for the agents making decisions. The LLM agent receives a compressed profile to emulate with the first task of filtering out amenities that are not relevant to that profile. The profile information is maintained but the filtering tasks are removed from memory for final process. The last step, forming the right side column, is for the agent to pick the sequence of locations out of the collection of plausible choices that remain. OpenStreetMap contributors 2025

The second component focuses on generating behavioral data by simulating decision making processes for agents initialized with the compressed profiles generated before. Each profile serves as the basis for determining a plausible subset of amenities and points of interest (POIs) relevant to the individual it represents.

The process begins by passing the compressed profile to a LLM, which is prompted to identify categories of amenities most applicable to the synthetic individual. The amenity selection is then refined through an additional LLM-based filtering step, ensuring that only the most relevant POIs are retained for subsequent planning. Although these agents are not formal multi-step reinforcement learners, they operate through a truncated action loop implemented via sequential LLM queries.

Concurrently, a query is submitted to the OpenStreetMap (OSM) application programming interface (API) to retrieve geographic data for amenities and POIs within a configurable radius, which is set to 20 km from a specified center point in this implementation. The OSM response includes names, coordinates, and categorical information for entities such as cafés, restaurants, libraries, and universities. To reduce redundant API calls and minimize resource consumption, the retrieved data is cached for reuse across multiple agents.

After these steps, the synthetic profile and the reduced amenity set are jointly provided to an LLM for further refinement. This stage, represented by the filter icon in Figure 3, emulates an agent action that prunes excess POIs from the dataset. This pruning is essential for reducing memory usage and minimizing the context window size required for downstream processing. Large context windows remain a practical limitation in LLM-based systems, and avoiding them significantly improves the usability of this approach.

The shortened amenity list and the synthetic profile are then combined in a final query that requests an ordered sequence of locations to visit. This step leverages the decision-making and behavior-emulation capabilities of LLMs within the domain of natural language reasoning to produce plausible daily activity patterns. The output is a structured list of locations that reflects the behavioral tendencies implied by the profile attributes. GPS coordinates for these locations are extracted into an array for export to subsequent stages or for independent use.

## 3.3 ROUTE AND WAYPOINT VISUALIZATION

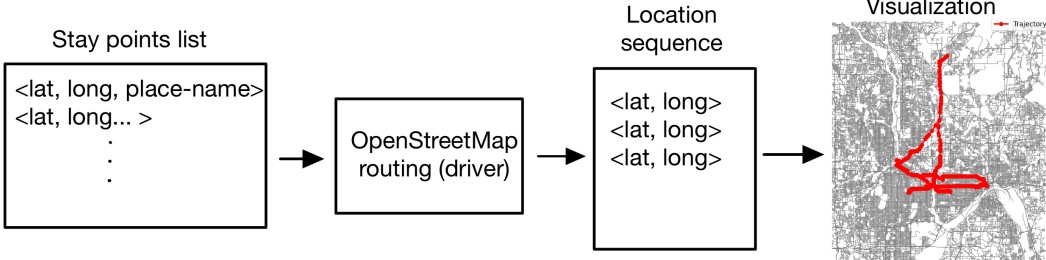

Figure 4: This section converts the previously generated stay points into a routed collection of waypoints reflecting a plausible mobility trace between the stay points. This process is simple though there are many alternatives to using OSM routing tools here.

The third component, illustrated in Figure 4, begins with a list of stay points that resemble those found in traditional travel surveys, but with greater specificity regarding the exact locations visited by the synthetic individual. The objective of this stage is to generate realistic routes connecting these locations, under the assumption that most people follow commonly suggested navigation paths. The resulting trajectories can be augmented with temporal information and, if desired, perturbed to introduce spatial noise that mimics the jitter often observed in location-based services.

To address the map matching challenge of aligning generated stay points with realistic road networks, this component utilizes OSM data OpenStreetMap contributors, 2025. Because the previous step provides a macroscopic sequence of GPS coordinates, these can be directly integrated with OSM routing services, or any other, to construct plausible paths along existing roads. This approach

ensures that the synthetic trajectories maintain geographic consistency while preserving the privacy guarantees established in earlier stages.

The final output of this component is a set of ordered waypoints representing the complete route for the day. These waypoints can be exported for visualization or further analysis, providing both a spatially accurate representation and a foundation for generating time-stamped mobility traces.

# 4 EARLY RESULTS

Data location privacy concerns make this approach difficult to evaluate directly for *en masse* comparisons. No previous methods in generating synthetic human mobility data cover the full range of what this approach produces while also using minimal input data. The overall outputs at this time do not have temporal components added in no other noise attributes that would actively reduce the distinguishability between one of these synthetic trajectories and a real one simply based on the unevenness of the appearance on a map visualization.

We chose to generate synthetic mobility traces starting with a center point of Minneapolis, Minnesota. There is extremely high matching of the synthetic trajectories to real road networks and variability in the chosen points of interest.

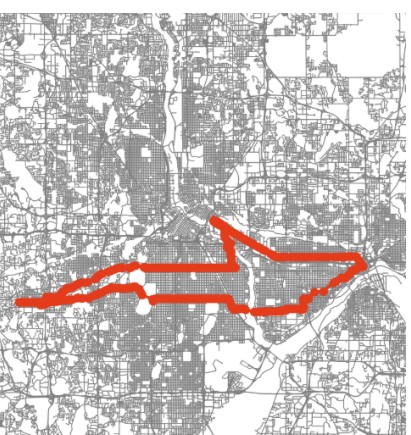

Figure 5: Working professional with strong coffee preferences and a dinner event in another city of Minnesota.

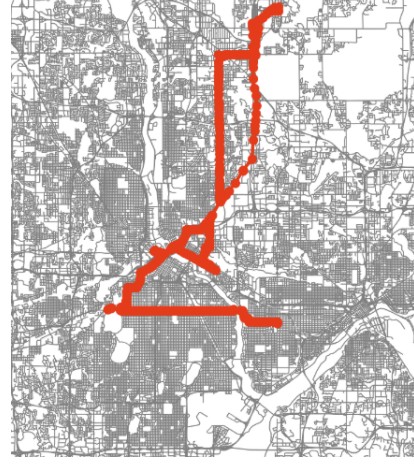

Figure 6: An (incomplete) trip to the Twin City suburbs

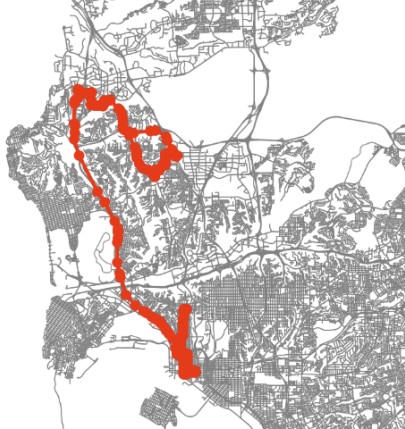

Figure 7: Northern San Diego, California multi-stop trip ending in Claremont.

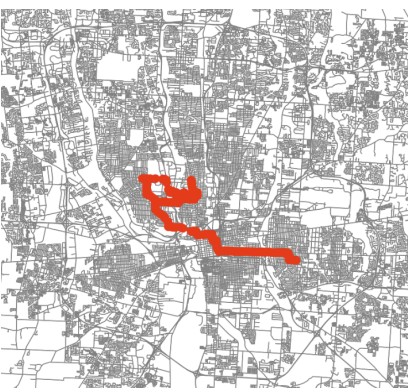

Figure 8: The path of a working professional in Columbus, Ohio

An example of the outputs from this approach can be best seen in Figure 5. This was a generated profile of a person that is a software engineer with a penchant for coffee. They go out of the way to get coffee at a chosen coffee shop before driving in to their work in St. Paul. At the end of the day, according to the agent, they had a dinner event.

In Figure 6 there is an example where the decision making agent chose too many destinations. The output contained 10 stops with several of them being coffee shops.

Figure 7 depicts an output from the approach when generated for another city, San Diego. Due to only requiring a small request to OSM to get the spatial information and visualization background for nearly any city, it is trivial to produce these trajectories for any other regions with rich OSM reporting.

In Figure 8, the generated route was for a doctor in Columbus, Ohio. The path takes them to a hospital from a suburb of Columbus which demonstrates the agentic decision making can succeed in arbitrary urban regions. However, the routing also had an additional coffee, lunch, and dinner stop. This route is not necessarily impossible, though, these stops seem unlikely given the locations selected are not close to the hospital.

Broadly, model biases and a minimal agent design affect the quality and realism of this approach negatively. More complex and iterative LLM agent approaches quickly increase the computational load; however, the general rate of production on a single desktop with the outlined approach was around 1000 novel profiles and trajectories per hour.

## 5 DISCUSSION

The method presented here is an ensemble approach to generate synthetic human mobility data. These early results present a compelling case for a thorough and scalable concept to capture both the nuances of agent based behavior pattern generation, supplied by generated profiles, and ground it with real spatial data when generating synthetic human mobility data.

Further study is needed to establish the limits of scaling this approach.

Several components of this approach have room for obvious improvements or optimizations that are either mentioned or implied throughout. These speak to the capacity this approach may have in future iterations given the realistic shapes and decisions present in the current outputs.

In the current construction of this approach, the most effective immediate use is for exploring counterfactual mobility behaviors and patterns. The agentic decision maker is highly tunable among other components, as previously mentioned, such that any assumptions included in the design or instructions will be reflected in the outputs in some way. This means that in the pursuit of establishing the most accurate design or instructions to capture broad ranges of behavior, perhaps even by using several different agents or configurations, the flaws may originate in the design as a result of an incomplete understanding of mobility behavior. Additionally, the use of these flawed understandings also allows for immediate testing. The remaining limitations are a lack of access to a ground truth dataset to compare against and ensuring the agent decisions are accurately reflecting the modified instructions in their processes.

Regional mobility patterns will likely require specific, focused modifications to this approach to achieve the desired performance levels. However, this opens up a specific avenue for broader mobility research acceleration. Researchers with privileged access to raw mobility data for specific regions can release versions of this approach with any additional curated publicly shareable data desired as a form of reliably generating synthetic mobility data for that region.

To summarize, we present a novel approach for generating synthetic human mobility data. Our approach generates synthetic mobility traces from artificially produced agent profiles and individualized, semantically meaningful agent decisions using limited data inputs. This approach works nearly instantly for any region with OSM coverage. However, it is not complete. A thorough analysis of the output distributions, and of the degree of control over them, will be vital to future developments.

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

## A  LLM Usage Disclosure

LLMs were used to aid in the simplification and polishing of the methods section. In the background section, LLMs were utilized for removing information that was no longer useful for establishing strong context.

