# OpenReview forum: "Towards a Comprehensive Solution for Generating Synthetic Human Mobility Data from Limited Input Data"
_ICLR.cc/2026/Conference — Submitted to ICLR 2026_

### Official Review · Reviewer_AWXB · 2025-10-27

**Soundness:** 2
**Presentation:** 1
**Contribution:** 1
**Rating:** 2
**Confidence:** 5

**Summary:**

This paper proposes a Data Minimal Framework designed to address the challenge of obtaining real human mobility data, which is difficult to acquire due to privacy concerns. The proposed framework generates synthetic mobility trajectories using only limited input data (a central point and OpenStreetMap data) without requiring additional model training. Its core methodology comprises three steps: 1. Using a LLM to create diverse synthetic agent profiles; 2. LLM agents generate sequences of stop points throughout the day based on their profiles and real facility data from OSM; 3. Connecting these stop points using OSM routing services to produce final trajectories matching the actual road network. Early results demonstrate high alignment between generated paths and the road network.

**Strengths:**

1. This framework requires only two external data sources and does not necessitate additional model training. This design avoids the use of real personal user data, thereby eliminating the risk of data leaks and protecting privacy.
2. The trajectories generated by this framework achieve an “extremely high” level of alignment with the actual road network, and through an agent-based decision-making process, ensure semantic consistency of the trajectories.
3. This method is location-independent. For any area with extensive OpenStreetMap coverage, generating tracks is straightforward and works.

**Weaknesses:**

1. This paper is incomplete, lacking sufficient detail to understand and reproduce the work. It fails to provide necessary experimental validation or comparisons with other methods.
2. The currently generated trajectory output lacks a time component, making it impossible to determine whether it is reasonable.
3. The proposed method cannot capture specific movement patterns and cannot modify or control alignment.
4. Generating trajectories using existing road networks and data points from OpenStreetMap certainly aligns closely with real-world road networks, but this is meaningless.

**Questions:**

Lack of sufficient model and methodological details.
What LLM model was used?

How was data processed?

What is the time efficiency?

What are the advantages over other methods?

---

### Official Review · Reviewer_BWSv · 2025-10-30

**Soundness:** 1
**Presentation:** 1
**Contribution:** 1
**Rating:** 0
**Confidence:** 5

**Summary:**

This paper proposes an ensemble-based approach for generating synthetic human mobility data. The authors present preliminary results suggesting a potentially scalable framework that captures both the nuances of agent-based behavioral pattern generation through generated user profiles and the grounding of these profiles in real spatial contexts when producing synthetic mobility traces.

**Strengths:**

The paper has been successfully submitted to ICLR.

**Weaknesses:**

1. This work appears to be an unfinished manuscript. It lacks any quantitative experiments, does not fill the expected page limit, includes roughly sketched figures, and even contains seemingly irrelevant content such as the unexplained “OpenStreetMap Contribution” at the end of line 215. **Submitting such an incomplete draft to a top-tier conference like ICLR is inappropriate and shows a lack of respect for the reviewers’ time and the venue’s standards.**

2. Methodologically, the proposed approach seems to be an oversimplified version of AgentMove with a few added visualizations. There is no substantial methodological novelty or technical contribution beyond existing frameworks.

[1] Feng, Jie, et al. "Agentmove: A large language model based agentic framework for zero-shot next location prediction." arXiv preprint arXiv:2408.13986 (2024).

**Questions:**

Please clarify and address the weaknesses mentioned above

---

> ### Author Response · Authors · 2025-11-23
>
> W2: To my understanding, the suggested paper, AgentMove, covers a next location prediction task, and this submission presents a framework for generating synthetic mobility data. Can you say more about the similarities you are concerned with?

---

### Official Review · Reviewer_ynDo · 2025-10-31

**Soundness:** 3
**Presentation:** 2
**Contribution:** 2
**Rating:** 6
**Confidence:** 3

**Summary:**

The paper proposes a data-minimal framework for generating synthetic human mobility trajectories from limited inputs. The method combines pre-trained large language models (LLMs) and open-source geospatial information (OpenStreetMap) to generate semantically coherent, spatiotemporally feasible, and privacy preserving mobility data without additional training. This work offers a promising solution to the real-world scarcity of mobility data caused by privacy constraints and is applicable across various scenarios.

**Strengths:**

1. “Agent profiling,” “agent decision making,” and “route visualization” form a complete task; this decomposition is easy to understand, and the LLM combined with a geographic information system can convert abstract attributes into sequences of concrete activities.
2. The data minimization principle enables simulation based only on region and population, protecting privacy while improving model scalability and generalization.

**Weaknesses:**

1. The flowchart is overly simplistic.
2. Without baseline comparisons, the four examples are unconvincing; please supply more, e.g., overlay multiple routes in one figure. Objective efficiency metrics are also missing, and the example images differ in size and are misaligned.
3. The paper states that LLM decisions are uninterpretable; can unreasonable decisions be detected or corrected afterward?
4. Only routes are simulated; temporal aspects such as commute and dwell times are ignored, limiting real-world utility.

**Questions:**

1. Could additional comparative examples be provided?
2. Could efficiency-metric comparisons against other baselines be added?
3. Is integration with temporal information achievable?
4. Is there any room to optimize the uninterpretable decisions?
5. Could the prompt templates and LLM versions be released to ensure reproducibility?

---

### Official Review · Reviewer_gYzo · 2025-11-02

**Soundness:** 1
**Presentation:** 2
**Contribution:** 1
**Rating:** 2
**Confidence:** 4

**Summary:**

The authors propose a "data-minimal" framework to generate synthetic human mobility trajectories without requiring any real-world trace data for training. The approach relies on two external, publicly available sources: a large language model (LLM) and OpenStreetMap (OSM). The method consists of three main components: Profile Generation,  Agent Decision-Making and Route and Waypoint Visualization.

**Strengths:**

1. The primary contribution is the introduction of a novel paradigm that leverages LLMs for behavioral simulation.
2. By completely avoiding the use of real individual mobility traces for training, it mitigates the risk of data leakage.

**Weaknesses:**

W1: Lack of quantitative evaluation and ground truth comparison: The authors fail to compare the statistical properties of the generated data (e.g., trip length distributions and activity patterns) with any real-world mobility datasets.
W2: Limited technical contribution compared with recent LLM-based human trajectory generation work, e.g.:
[1] Wang, J., Jiang, R., Yang, C., Wu, Z., Onizuka, M., Shibasaki, R., ... & Xiao, C. (2024). Large language models as urban residents: An llm agent framework for personal mobility generation. Advances in Neural Information Processing Systems, 37, 124547-124574.
[2] Du, Y., Feng, J., Zhao, J., Yuan, J., & Li, Y. (2024). TrajAgent: An LLM-based Agent Framework for Automated Trajectory Modeling via Collaboration of Large and Small Models. arXiv preprint arXiv:2410.20445.
[3] Gao, C., Lan, X., Li, N., Yuan, Y., Ding, J., Zhou, Z., ... & Li, Y. (2024). Large language models empowered agent-based modeling and simulation: A survey and perspectives. Humanities and Social Sciences Communications, 11(1), 1-24.
W3: Absence of check-in points or stay points in Figures 5–8: These figures fail to represent users' activity trajectories.
W4: Irrelevant and incoherent background section: The content in the background section is not closely relevant to the study’s core focus, and there is no clear coherence between its paragraphs.

**Questions:**

Q1: Why no quantitative evaluation?
Q2: Discussion on technical contribution considering these existing work in W2.

---

### Meta-Review · Area_Chair_ZbNT · 2026-01-07

**Summary:**

This paper proposes a data-minimal framework for generating synthetic human mobility trajectories using only pre-trained LLMs and OpenStreetMap, without training on real mobility data. Reviewers agree that the motivation, privacy-preserving and scalable mobility generation is important and timely, and that LLM-based agentic simulation is a promising direction. However, the majority of reviewers find the submission incomplete for ICLR, citing the absence of quantitative evaluation, limited demonstrated novelty over existing LLM-based mobility frameworks, missing temporal modeling, and weak presentation quality. Overall, the work is viewed as an early prototype rather than a mature research contribution.

**Reviewer Concerns:**

The rebuttal partially clarifies the distinction between this work and AgentMove by emphasizing end-to-end synthetic data generation rather than next-location prediction. However, the most critical concerns remain unresolved, and the author did not reply to most of the reviews. Reviewers consistently note the lack of quantitative evaluation or comparison to real-world mobility data or existing baselines, making it impossible to assess realism or utility. Concerns about limited technical contribution relative to recent LLM-based agent frameworks, the absence of temporal components such as dwell times and schedules, presentation issues, missing methodological details, and insufficient reproducibility information are also mentioned by reviewers, but not discussed in the rebuttal.

**Reviewer Scores:**

Reviewer gYzo would likely remain at reject. Reviewer ynDo would likely remain borderline, with a possibility of a lower score. Reviewer BWSv would likely remain at strong reject. Reviewer AWXB would likely remain at reject due to unresolved concerns regarding soundness, novelty, and evaluation.

---

### Decision · Program_Chairs · 2026-01-26

Reject